# Teenage pregnancy and timing of first marriage in Cameroon—What has changed over the last three decades, and what are the implications?

Jean Christophe Fotso[1]*, John G. Cleland[2,3], Berienis Muki[1‡], Elihou Adje Olaïtan[1‡], Josiane Ngo Mayack[4]

1 EVIHDAF, Yaoundé, Cameroon, 2 Department of Population Health, London School of Hygiene & Tropical Medicine, London, United Kingdom, 3 International Advisory Board Member, EVIHDAF, Yaoundé, Cameroon, 4 Individual Consultant, Yaoundé, Cameroon

☯ These authors contributed equally to this work.
‡ BM and EAO also contributed equally to this work.
* jcfotso@evihdaf.com

**Data Availability Statement:** The data are from a publicly available source, the Demographic and Health Surveys (DHS). Here are the direct URLs to access the publicly available datasets for

## Abstract

### Background

The consequences of teenage childbearing on the health of mothers and children, and on girls' schooling have been documented in many studies. The objectives of this study are to: 1) examine trends and differentials in teenage motherhood in Cameroon, with a distinction between premarital and marital teenage pregnancy; and 2) investigate trends and differentials in the length of time to marriage following a premarital teenage pregnancy.

### Methods

We use data from five demographic and health surveys (DHS) conducted in Cameroon between 1991 and 2018. Teenage pregnancy, defined as first pregnancy occurring before the age of 20 years, is recode as a trichotomous variable (0 = No teenage pregnancy; 1 = marital teenage pregnancy; 2 = premarital teenage pregnancy). Time from first premarital teenage pregnancy to first marriage is analyzed as a continuous variable.

### Results

The percentage of women who experienced a marital teenage pregnancy declined from 39.6% to 26.4% between 1991 and 2018. After an initial drop between 1991 and 2004, premarital teenage pregnancy stabilized at about 25%. Women with intermediate levels of schooling were more likely to experience a premarital pregnancy than those with no schooling or higher secondary/tertiary education. The median length of time to first marriage following a premarital teenage pregnancy rose from 16 months in 1991 to 45 months in 2018. Further analysis suggests that marriage may be a more severe barrier to continued schooling than motherhood and that the desire to continue schooling is an important reason for postponing marriage for women who have given birth.

Cameroon: -Year 2018: https://dhsprogram.com/methodology/survey/survey-display-511.cfm -Year 2011: https://dhsprogram.com/methodology/survey/survey-display-337.cfm -Year 2004: https://dhsprogram.com/methodology/survey/survey-display-232.cfm -Year 1998: https://dhsprogram.com/methodology/survey/survey-display-101.cfm -Year 1991: https://dhsprogram.com/methodology/survey/survey-display-38.cfm One would need to request access by sending an email to archive@dhsprogram.com, specifying the countries of interest.

**Funding:** The authors received no specific funding for this work.

**Competing interests:** The authors have declared that no competing interests exist.

## Conclusion and recommendations

Besides strengthening interventions to curb adolescent pregnancy, efforts should be made to support families, communities and schools to help adolescent mothers return to school, prevent future unintended pregnancies, and delay further family formation. Accessibility to youth-friendly FP/RH services should be addressed.

## Introduction

In the developing world, teenage pregnancy and early marriage continue to derail girls' health and socioeconomic development prospects, and to undermine efforts aimed at lifting families and communities out of poverty [1–4]. The consequences of teenage childbearing on the health of mothers and children have been documented in many studies across all regions of the world. Teenage mothers are at higher risk of obstetric complications such as incontinence from obstetric fistulae, eclampsia, post-partum hemorrhage, and sepsis, making pregnancy and childbirth complications the leading cause of death among girls aged 15–19 years globally. Further, babies born to mothers under 20 years of age face higher risks of low birth weight, preterm delivery and severe neonatal conditions [1, 2, 5–10]. There is also compelling evidence that teenage marriage and motherhood significantly contribute to high school dropout and lower educational attainment [11–14].

Despite rising age at first marriage, teenage fertility remains high in sub-Saharan Africa. Trend data show a general decline, but progress has been slow and patchy [5]. The highest adolescent birth rate is found in West and Central Africa with 108 births per 1,000 adolescents, compared to 95 births per 1,000 adolescents in East and Southern Africa [15]. Extensive literature suggests that early marriage and teenage childbearing in West and Central Africa result from a broad array of historical, cultural, and socioeconomic factors, including pronatalist beliefs and socio-cultural norms in favor of child marriage in some communities, and low contraceptive use [16–18]. Polygyny, a common practice in the region, makes young girls attractive co-wives [19, 20]. There are numerous publications on correlates of early marriage and motherhood in Africa and globally, which almost all point to rural, ill-educated girls from poor households being particularly at risk [21–26]. While these studies have provided valuable insights into teenage motherhood, none sought to separate the premarital and the within-union teenage childbearing, yet each component may have different drivers and lead to different implications.

The length of time from a premarital teenage pregnancy to a first marriage has received little attention, limiting our understanding of the relative importance of teenage motherhood versus early marriage on adolescent well-being. The consequences of premarital childbearing in Cameroon for the timing of subsequent marriage were examined using the 1991 DHS [27]. Single mothers were more likely to marry or cohabit than their childless peers in the first two years after birth, presumably to the biological father, but less likely at longer durations. Further evidence comes from a multi-country analysis which included the 2011 Cameroon DHS. Among all parous women in this survey, 20% were still single at time of first birth and an additional 8% married while pregnant with the first birth [28]. In a more recent multi-country analysis, no effect of premarital childbearing on subsequent marriage dissolution or polygynous marriage was found in Cameroon [29]. None of these studies, however, focused on the adolescence window.

In Cameroon, until 2016 the legal minimum age of marriage was 15 years for girls subject to parental authorization, and 18 years for boys. In July 2016, a revised article of the civil code set it to 18 years for males and females. Besides the Ministry of Public Health, the health and education needs of adolescents are also taken care of by the Ministries of Youth and Civic Education, Basic Education, Secondary Education, Higher Education, Social Affairs, Women's Empowerment and the Family, and by a few organizations and programs, some of which have established adolescent-friendly reproductive health centers. However, the services provided are generally poor due to lack of infrastructure, and are underused by young people due to a lack of communication and lack of qualified personnel. Furthermore, there is no synergy of actions or exchange platform between these institutions [30]. In the education system, student pregnancy is addressed by a 1980 ministerial circular which provides that a girl should be immediately suspended from school if she is found to be pregnant, and that she may resume classes after giving birth [31]. However, most of the girls who are sanctioned do not return to school. In 2019, Cameroon joined the "*Sahel Women Empowerment and Demographic Dividend*" project (SWEDD-2), and is using this opportunity to conduct a study that will enable the various actors in the system to grasp all the contours of the problem of continuation of schooling for pregnant students, in order to provide an appropriate response [32].

Against this backdrop, this study aims to improve our understanding of the patterns of, and disparities in teenage childbearing in Cameroon over the last three decades, in order to inform the design of strategies that can accelerate the prevention of adolescent pregnancy. Its objectives are to: 1) examine trends and differentials in teenage motherhood in Cameroon, with a distinction between premarital and marital teenage pregnancy; and 2) investigate trends and differentials in the length of time to marriage following a premarital teenage pregnancy.

## Materials and methods

We use secondary data from the five rounds of DHS conducted in Cameroon between 1991 and 2018. DHS are nationally-representative surveys organized in developing countries to generate data on population, health and reproductive health, along with various information on the demographic and socioeconomic characteristics of households and their members. The individual working datasets for the years 1991 and 1998 are derived from the merged women and wealth files. For the years 2004, 2011 and 2018, the working datasets are retrieved from the corresponding women files, since they contain the wealth variable. All five datasets are merged to form the final working dataset. The analysis is restricted to women 20–24, since they have been fully exposed to the risk of teenage pregnancy and their young age should reduce recall error. **Ethical clearance was not needed for the study**.

### Outcome variables

**Teenage pregnancy** is defined as first pregnancy occurring before the age of 20 years (or 240 months). Age at first pregnancy and age at first marriage in months, are calculated using the century month code (CMC) versions of date of first birth, date of first marriage, and date of interview. For women who were pregnant for the first time, their age at first pregnancy is calculated by subtracting the number of months of pregnancy from the date of interview. Teenage pregnancy is classified as premarital or marital, by comparing the age at first pregnancy and the age at first union. There were 170 women (out of 9,736) with age at first pregnancy the same as age at first marriage. They are classified as having a within-marriage teenage pregnancy, with the assumption that marriage is likely to have been planned before the pregnancy. The teenage pregnancy status is defined as follows:

- No teenage pregnancy (coded 0), if age at first pregnancy > = 240 (months)

- Marital teenage pregnancy (coded 1), if age at first pregnancy < 240 **and** age at first pregnancy > = age at first marriage

- Premarital teenage pregnancy (coded 2), if age at first pregnancy < 240 **and** age at first pregnancy < age at first marriage

In the absence of a complete pregnancy history this status is derived from pregnancies that ended in a live birth (or current pregnancies), and does not consider miscarriages or abortions.

**Time from first premarital teenage pregnancy to first marriage,** in months, is obtained by subtracting age at first premarital teenage pregnancy from age at first marriage. The analysis is restricted to women age 20–24 who had a first premarital teenage pregnancy.

## Covariates

The key covariates for this study are the survey year, place of residence, and region of residence. The regions of Adamaoua, North and Far-North form the Northern zone. Islam is the main religion, the level of education is relatively low, and child marriage is very prevalent. The Central, East and South provinces are home to the Beti ethnic group. Sexual norms are more liberal among this group, with premarital fertility tolerated and even valued as proof of maturity. Residents of the North-West and the West provinces share similar values and norms regarding sexuality, marriage and childbirth, which are viewed as largely restrictive. Finally, populations in the Littoral and South-West regions have values and norms that tend to be relatively permissive [33].

The asset-based household wealth, which is a good proxy for economic status, is recoded as tertiles (lowest, middle, highest). With the caution of possible two-way relationship between education and teenage pregnancy in mind, we include education in the analysis. It is recoded as a four-category variable: none, primary, lower secondary (7–10 years of education), higher secondary/tertiary (11+ years of education. We also include religion though it is not uncorrelated with region of residence, with the following categories: Catholics, Other Christians, Muslims, and No religion/Other.

## Data analysis

The analyses, performed using STATA software, are weighted with the ***svyset*** command. We first perform descriptive analyses on and use multinomial logistic regression with the category "N*o teenage pregnancy*" as the reference, to quantify the "net" differentials in marital teenage pregnancy and premarital teenage pregnancy. For the analysis of length of time from a first premarital teenage pregnancy to first marriage, we first generate descriptive estimates using Kaplan-Meier survival analysis. Women who were not married by date of the survey are right-censored, and their observation time is the number of months between their first pregnancy and the survey date. This descriptive analysis is followed by bivariate and multivariate Cox proportional hazards regression models. We tested the assumption inherent to Cox proportional hazards regression, and adjusted the models accordingly, using STATA's time varying covariate (tvc) command.

## Results

**Table 1** shows the distribution of the sample of women 20–24. **Fig 1** shows that the percentage of women who experienced a marital teenage pregnancy declined steeply from 39.6% to 26.4%

**Table 1. Sample characteristics for, and multinomial logistic regression of teenage pregnancy; Cameroon 1991–2018 DHS.**

| | Sample characteristics | | Multivariate, multinomial regression | | | |
| | | | Marital | | Pre-marital | |
| | | | versus | | versus | |
| | | | No teenage motherhood | | No teenage motherhood | |
| | N[a] | %[b] | RRR | P-value | RRR | P-value |
|---|---|---|---|---|---|---|
| Survey year | | | | | | |
| 1991 | 789 | 8.1 | 1.89 | 0.000 | 1.40 | 0.003 |
| 1998 | 1,153 | 11.8 | 1.13 | 0.242 | 0.85 | 0.094 |
| 2004 | 2,214 | 22.7 | 1.39 | 0.000 | 0.92 | 0.302 |
| 2011 | 3,118 | 32.0 | 1.28 | 0.002 | 0.93 | 0.299 |
| 2018 | 2,462 | 25.3 | 1.00 | | 1.00 | |
| Residence | | | | | | |
| Douala/Yaoundé | 2,184 | 23.2 | 1.00 | | 1.00 | |
| Other urban | 3,193 | 31.9 | 1.30 | 0.005 | 1.53 | 0.000 |
| Rural | 4,359 | 45.0 | 1.69 | 0.000 | 2.03 | 0.000 |
| Region | | | | | | |
| Adamaoua/North/Far North | 2,553 | 29.2 | 1.40 | 0.001 | 0.52 | 0.000 |
| Centre/East/South | 3,368 | 30.2 | 2.06 | 0.000 | 1.92 | 0.000 |
| Littoral/South West | 2,142 | 21.0 | 1.00 | | 1.00 | |
| North West/West | 1,673 | 19.6 | 1.15 | 0.168 | 0.79 | 0.010 |
| Household wealth | | | | | | |
| Lowest | 2,675 | 29.4 | 1.66 | 0.000 | 1.78 | 0.000 |
| Middle | 3,180 | 31.2 | 1.44 | 0.000 | 1.57 | 0.000 |
| Highest | 3,881 | 39.4 | 1.00 | | 1.00 | |
| Education | | | | | | |
| None | 1,512 | 17.6 | 2.97 | 0.000 | 2.19 | 0.000 |
| Primary | 2,970 | 30.3 | 1.85 | 0.000 | 1.31 | 0.000 |
| Low Secondary | 2,956 | 28.6 | 1.00 | | 1.00 | |
| High Secondary+ | 2,298 | 23.5 | 0.13 | 0.000 | 0.25 | 0.000 |
| Religion | | | | | | |
| Catholics | 3,890 | 39.0 | 1.00 | | 1.00 | |
| Other Christians | 3,545 | 35.2 | 0.98 | 0.746 | 0.98 | 0.775 |
| Muslims | 1,791 | 19.9 | 1.77 | 0.000 | 0.91 | 0.340 |
| No religion/Other | 510 | 5.9 | 1.10 | 0.474 | 0.84 | 0.185 |
| N | 9,736 | NA | NA | NA | NA | NA |

between 1991 and 2018. After an initial drop between 1991 and 2004, premarital teenage pregnancy changed little. In 2018, the probability of a premarital teenage pregnancy was the same as a marital pregnancy. First premarital pregnancies for which births occurred within marriage represented 33% of all first premarital teenage pregnancies on average, with a clear downward trend'.

Across the years and the equity dimensions (residence, wealth and education), differences were substantial for marital pregnancy, and narrow to negligible or counterintuitive for premarital pregnancy (**Fig 2**). On average across all surveys, rural women were about 2.7 times more likely than Douala/Yaoundé residents to be pregnant within marriage (44.1% versus 16.3%), and only 1.4 times more likely to experience premarital pregnancy (29.9% versus 21.5%). Larger disparities in within-union pregnancy compared to premarital pregnancy are also observed with regard to household wealth. The differences by education were even more

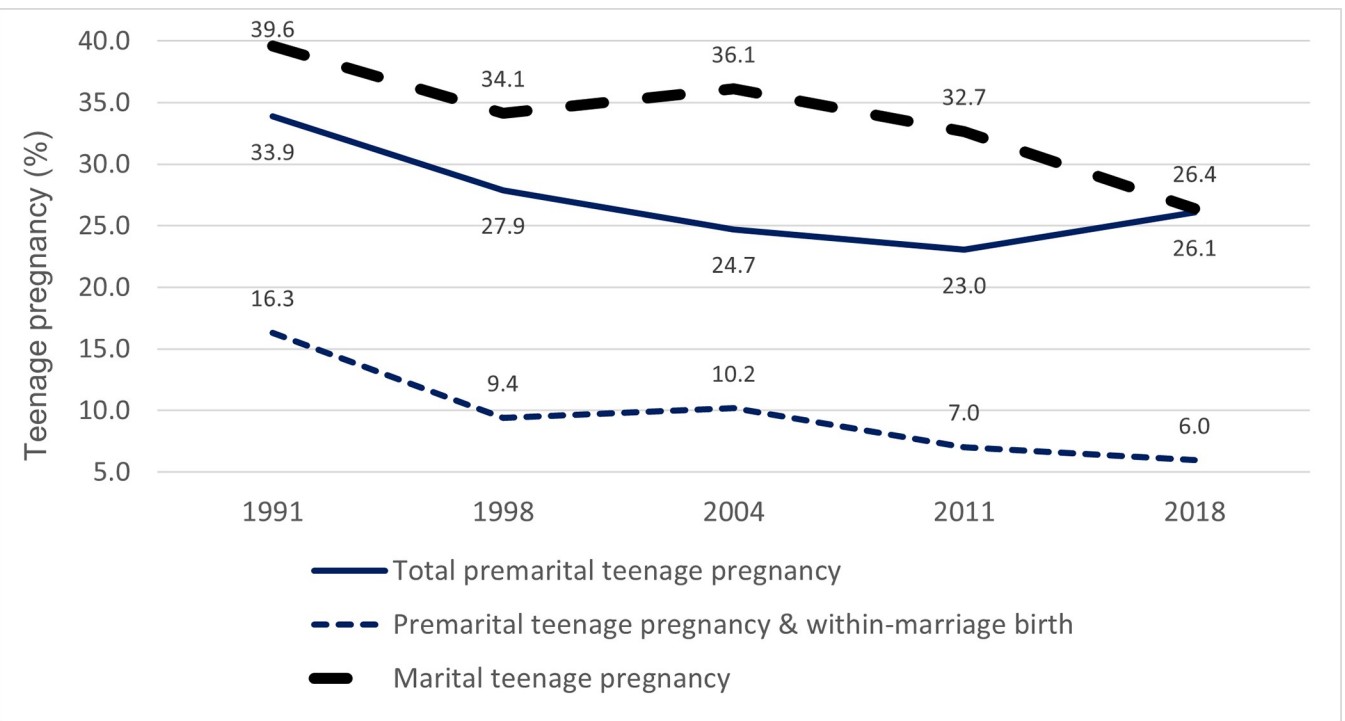

**Fig 1. Trends in marital and premarital teenage pregnancy among women 20–24, Cameroon 1991–2018 DHS.**

pronounced. While women with no education were on average 2.3 times more likely than their peers with lower secondary education, to experience a marital teenage pregnancy, the likelihood of premarital pregnancy increased with level of education until the highest level of education: 23.6% among non-educated, 29.8% among women with primary education, and 32.2% among women with lower secondary education, on average. Overall, women with inter-mediate levels of schooling (primary or lower secondary) recorded consistently higher probabilities of having a premarital pregnancy than those with no schooling or with the highest level of schooling. It should be noted that premarital pregnancy increased between 2011 and 2018 for those with lower or higher secondary schooling.

**Table 1** shows the multivariate analysis of differentials in marital and premarital teenage motherhood. The top panel shows trends between 1991 and 2018. Even after adjustment for socio-economic factors, the probability of marital pregnancy fell. Premarital pregnancy also fell after 1991 but remained essentially unchanged since 2004. Socio-economic and residential disparities in marital pregnancy remained substantial, but unlike in **Fig 2**, premarital teenage pregnancy also showed substantial disparities. The relative risk ratio (RRR) for rural residents, compared to Douala/Yaoundé residents, was 1.69 (p = 0.000) for marital, and 2.03 (p = 0.000) for premarital pregnancy versus no pregnancy. With regard to household wealth, comparing women from the low economic status to those from high economic status, RRR estimate was 1.66 (p = 0.000) for marital, and 1.78 (p = 0.000) for premarital pregnancy. Comparable RRRs across the two outcomes were also observed with regard to education. Regional differences were pronounced. Compared with the Littoral/South West zone, the RRRs for both marital and premarital pregnancy were twice as high in the Centre/East/South zone. Marital pregnancy in the Adamaoua/North/Far North zone was higher than the reference zone but premarital pregnancy much lower. As for religion only one significant result is apparent: Muslims have a higher probability of marital pregnancy than women of other religions.

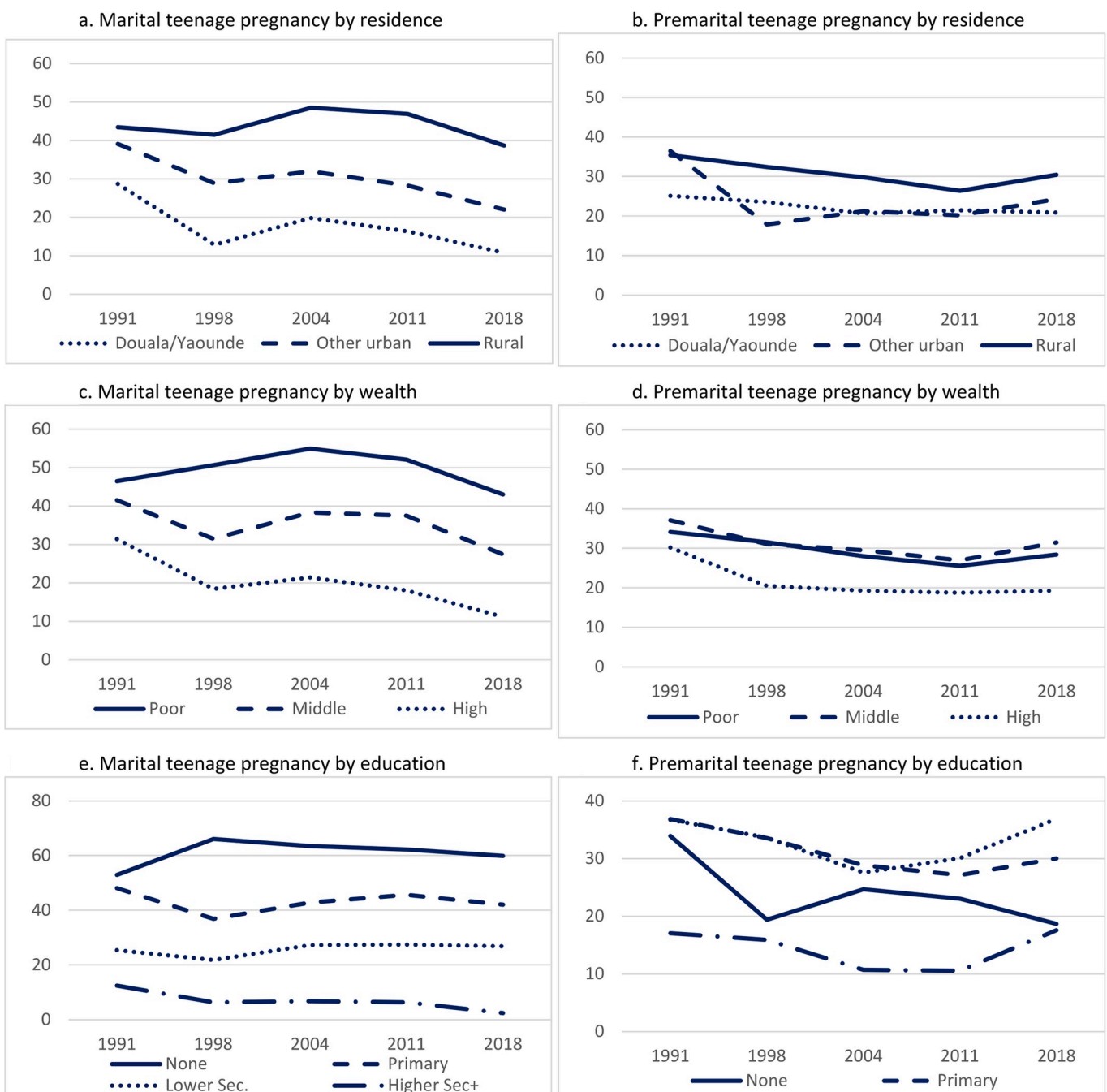

**Fig 2. Socio-economic differentials in marital and premarital teenage pregnancy among women 20–24, Cameroon 1991–2018 DHS.**

The median length of time to first marriage following a premarital teenage pregnancy rose gradually from 16 months in 1991 to 33 months during the period 1998–2011 (though there was a drop to 22 months in 2004), and further increased by to 45 months in 2018 (**Fig 3**). Compared to Douala/Yaoundé, the median length of time was shorter by 6 months in other urban areas, and by 20 months in rural areas. Differences by household wealth also appear strong, with median length of time longer by 16 months among women from middle wealth group, and longer by 24 months among those in the highest group, compared to the lowest wealth

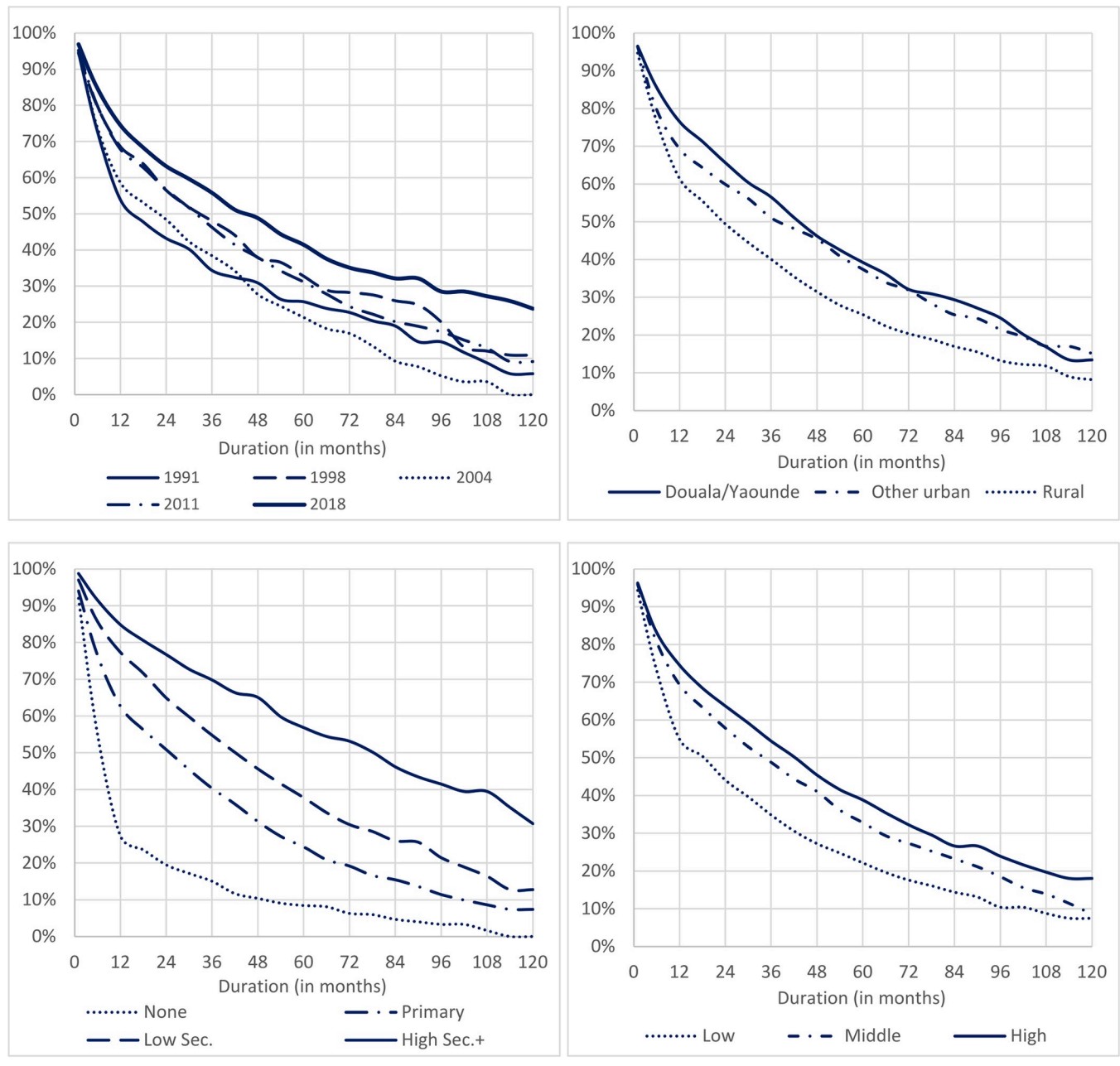

**Fig 3. Proportion of women 20–24 remaining unmarried after a first premarital teenage pregnancy, Cameroon 1991–2018 DHS.**

group. The differences by education appear even stronger, with median duration of 7 months among non-educated women, 26 months among women with primary education, 43 months among those with lower secondary education, and 79 months among women with higher secondary or tertiary education.

In **Table 2** we present the results of the Cox regression. Following a premarital pregnancy, the hazard of entry into marriage fell markedly between 2004 and 2011 and more gently between 2011 and 2018. The hazard of entry into first union was 50% higher for rural women than for residents of Douala/Yaoundé (p = 0.000) in the bivariate model, but the difference did not reach statistical significance in the multivariate results. Likewise, there was gradient along

**Table 2. Cox regression of the determinants of timing of first marriage following a first premarital teenage pregnancy, Cameroon 1991–2018 DHS.**

| | Bivariate model | | Multivariate model | |
|---|---|---|---|---|
| | Hazard Ratio | P-value | Hazard Ratio | P-value |
| Survey year | | | | |
| 1991 | 1.70 | 0.000 | 1.38 | 0.001 |
| 1998 | 1.28 | 0.004 | 1.21 | 0.032 |
| 2004 | 1.81 | 0.000 | 1.52 | 0.000 |
| 2011 | 1.35 | 0.000 | 1.14 | 0.054 |
| 2018 | 1.00 | | 1.00 | |
| Residence | | | | |
| Douala/Yaoundé | 1.00 | | 1.00 | |
| Other urban | 1.11 | 0.178 | 0.91 | 0.283 |
| Rural | 1.50 | 0.000 | 1.09 | 0.352 |
| Region | | | | |
| Adamaoua/North/Far North | 3.35 | 0.000 | 2.30 | 0.000 |
| Centre/East/South | 1.13 | 0.067 | 1.18 | 0.018 |
| Littoral/South West | 1.00 | | 1.00 | |
| North West/West | 1.26 | 0.005 | 1.14 | 0.158 |
| Household wealth | | | | |
| Lowest | 1.64 | 0.000 | 0.99 | 0.897 |
| Middle | 1.17 | 0.006 | 0.98 | 0.755 |
| Highest | 1.00 | | 1.00 | |
| Education | | | | |
| None | 3.02 | 0.000 | 1.58 | 0.000 |
| Primary | 1.48 | 0.000 | 1.45 | 0.000 |
| Lower secondary | 1.00 | | 1.00 | |
| Higher secondary+ | 0.58 | 0.000 | 0.54 | 0.000 |
| Religion | | | | |
| Catholics | 1.00 | | 1.00 | |
| Other Christians | 0.88 | 0.019 | 0.85 | 0.002 |
| Muslims | 2.20 | 0.000 | 1.29 | 0.002 |
| No religion/Other | 1.95 | 0.000 | 1.40 | 0.002 |

the wealth status categories in the bivariate model, but the multivariate results showed no difference by wealth group. The multivariate results indicate that though religion and region influenced hazard of marriage following teenage pregnancy, the strongest association was by far with education.

Because of the very strong association between education and the marriage hazard following a premarital pregnancy, we explored the relationship further by assessing current enrolment in school or college among women aged 20–24 in the 2018 DHS. School/college attendance was 66.2% among single women with no child, 34.5% among single women with a child, 16.7% among the small minority of married women with no child, and 5.6% among married women with a child. Single women with a child were therefore 7 times more likely to be in school than married women with a child, and twice as likely than married women with no child.

## Discussion

In this paper we assessed trends in teenage reproduction in Cameroon over a period of nearly three decades, using data from five DHSs. We made a distinction between conceptions that

occurred before and after marriage because nearly all marital first pregnancies in Africa are welcome whereas a large fraction of premarital pregnancies is unintended [34]. There is also evidence in Africa, though not for Cameroon, that single motherhood may have negative effects for the child and the mother [29]. In addition, appropriate policy responses differ. The most effective way to combat teenage marital pregnancy is to discourage early marriage, while premarital pregnancies can best be addressed by sexual and reproductive health education combined with easy access to contraceptives [35, 36].

We acknowledge the main limitation of the analysis, namely the need to restrict attention to pregnancies that ended in a live birth, with a small contribution from current pregnancies, because of the absence of data on abortion and miscarriages. The validity of the results also depends on the ability of women to recall dates We minimized recall problems by focusing on the testimony of young women aged 20–24 and have confidence in our estimates, with the exception of the 1998 survey which yielded fertility data of low quality [37].

Much changed in Cameroon between 1991 and 2018. In particular, a massive expansion of education took place. In 1991, only 38.5% of young women in their early 20s had secondary or higher schooling. By 2018, this proportion had risen to 64%. In an equally pronounced and related change, teenage marriage fell: in 2018, 19% of women aged 15–19 were married compared with 41% in 1991.

As use of effective contraception early in marriage is very low in Cameroon, the decline in teenage marriage guarantees a fall in teenage marital pregnancy; between 1991 and 2018, the probability of such a pregnancy dropped by one-third. As shown in Fig 2, wide socio-economic disparities in marital pregnancy persisted over the period. Differences by education were particularly pronounced but interpretation is complicated by the fact that marriage is likely to cause school drop-out [13].

The multivariate analysis of teenage marital pregnancy revealed strong cultural influences, consistent with qualitative evidence that attitudes to early marriage vary widely between ethnic groups [18]. Even after adjustment, women living in the Centre/East/South zone and Muslim women were at greater risk of pregnancy than others. But the most important conclusion is that the fall in risk of marital pregnancy since 1991 persists after adjustment. Thus, rising levels of education and urbanization cannot fully account for the decline in teenage marriage and marital pregnancy.

The direct determinants of teenage premarital pregnancy are age at sexual debut, age at marriage, and use of contraception. Abortion does not prevent pregnancy, but it does prevent live births. Median age at first sex, as reported by women aged 20–24, rose from 16.4 in 1991 to 17.5 in 2018. Additional downward pressure on pregnancy comes from increased use of modern contraception, mainly condoms, which rose from 3.5% in 1991 among sexually active single teenagers to about 50% in 2004, with no further increase thereafter. Specialized studies suggest that abortion among young women in Cameroon is common though no trend data are available [38, 39]. Offsetting the negative influences of these factors on the risk of a premarital pregnancy is the decline in teenage marriage which increases years of exposure.

The net effect of this interplay of factors was a sharp reduction in pregnancy risk between 1991 and 2004, in line with the increase in contraceptive use. Since 2004, pregnancy risk stabilized with the suggestion of a slight increase between 2011 and 2018. According to the 2018 data, the probability of a marital and a premarital pregnancy is now identical at 26%. Declines have also been documented for other African countries, including Burkina Faso, Senegal and Togo [40].

Disaggregated trends in the proportion having a premarital pregnancy showed little difference over the entire period of 1991 to 2018 between rural and urban women and between poor and rich. Proportions with a premarital teenage pregnancy were consistently higher for

women with intermediate levels of schooling (primary and lower secondary) and consistently lower for those with higher secondary or tertiary education. These results, though unexpected, are descriptively valid, but the major explanation is that uneducated women marry much earlier and thus have reduced exposure to premarital conception. Only in the highest education group, does effective fertility regulation offset greater exposure, though the probability of pregnancy in this group appears to have increased since 2011.

When the relative risk ratios of experiencing a premarital teenage pregnancy versus no marital or premarital teenage pregnancy were assessed by multinomial regression, the expected relationships were found, with much higher ratios among the rural than the urban and the poor than the rich. Even after adjustment for other covariates, very strong effects of region were apparent. Compared with the Littoral/South West zone, risk ratios were almost double in the Centre/East/South zone but half in Adamaoua/North/Far North. This result is consistent with ethnographic evidence that the Bulu-Beti-Fang ethnic groups living mainly in the zone with the highest risk of premarital pregnancy accept premarital sex and tolerate premarital births [41]. Education had by far the strongest associations with risk ratios but interpretation is difficult, because of bi-directional causality. School attendance may be a protective factor but pregnancy may truncate schooling. In Cameroon, as elsewhere in Africa, pregnancy often results in school-drop-out particularly during the first four grades of secondary school [12, 42]

Because of the possible negative consequences of single motherhood, we assessed the delay between a premarital pregnancy and marriage or cohabitation. A remarkable picture emerged. Between 1991 and 2018, the percentage who married before the birth of the child dropped from 16% to 6% and the median gap between pregnancy and marriage widened from 16 to 45 months. Education had by far the strongest association with marriage timing.

What could be the explanation for the trend for delayed marriage following premarital pregnancy and for the association with education? Is that men have are increasingly reluctant to acknowledge the child or face growing barriers in raising the traditional bride price, combined with disadvantage in the marriage market for single mothers, as suggested by Calves [27]. Or is it a choice by young women to forego marriage in order to continue their education as suggested by Smith-Greenaway and Clark [28]? We investigated further by examining current school/college attendance in our 2018 sample of women aged 20–24. Because of delayed entry to school and grade repetition, extension of secondary schooling beyond age 20 is common in Cameroon; 28% of all women were attending school or college, with striking differences by marital status. Interpretation must be cautious but it does appear that marriage may be a more severe barrier to continued schooling than motherhood and that the desire to continue schooling is an important reason for postponing marriage for women who have given birth.

## Conclusions and recommendations

This study has shown that despite progress in the last three decades, teenage pregnancy is experienced by half of young women and remains a concern in Cameroon. Curbing adolescent pregnancy and supporting teenage mothers to stay in school and delay further family formation should regain priority in efforts to accelerate countries' progress towards the achievement of health and development goals. While the Cameroon government should continue to enact laws against child/early marriage, with a focus on regions where this custom is most common, and promote girl's education, efforts should be made to support families, communities and schools to help adolescent mothers return to school and prevent future unintended pregnancies. Adolescent and youth sexual and reproductive health should more strongly than in the past be positioned as a priority not only within the health system, but also in education sector.

Finally, reproductive health programs should strengthen and scale up interventions to address accessibility and acceptability of services tailored to the needs of adolescents and youth. This requires, among other things, training of service providers.

## Supporting information

**S1 Fig. Trends in within-marriage and premarital teenage pregnancy among women 20–24, Cameroon 1991–2018 DHS.**
(XLSX)

**S2 Fig. Trends and socio-economic differentials in marital and premarital teenage motherhood.**
(XLSX)

**S3 Fig. Proportion of women 20–24 remaining unmarried after a first premarital teenage pregnancy, Cameroon 1991–2018 DHS.**
(XLSX)

## Acknowledgments

The authors would like to thank M. Gansaonre Rabi Joel, PhD Candidate at the University of Laval, Canada, for his support with the survival analyses. He is affiliated with *Institut Supérieur des Sciences de la Population* (ISSP) at the University of Ouagadougou, Burkina Faso.

## Author Contributions

**Conceptualization:** Jean Christophe Fotso, John G. Cleland.

**Data curation:** Elihou Adje Olaïtan.

**Formal analysis:** Jean Christophe Fotso, Elihou Adje Olaïtan.

**Writing – original draft:** Jean Christophe Fotso, John G. Cleland.

**Writing – review & editing:** John G. Cleland, Berienis Muki, Josiane Ngo Mayack.

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
