## [Decision Letter · Decision Letter 0]

16 Aug 2022

PONE-D-22-18046Teenage pregnancy and timing of first marriage in Cameroon - What has changed over the last three decades, and what are the implications?PLOS ONE

Dear Dr. Fotso,

Thank you for submitting your manuscript to PLOS ONE. After careful consideration, we feel that it has merit but does not fully meet PLOS ONE’s publication criteria as it currently stands. Therefore, we invite you to submit a revised version of the manuscript that addresses the points raised during the review process.

In submitting your revised manuscript, make sure you adequately address all the comments by the two reviewers, which are all critical to acceptance of your manuscript.

We look forward to receiving your revised manuscript.

Kind regards,

Grant Murewanhema, MD

Academic Editor

PLOS ONE

Journal Requirements:

Additional Editor Comments:

This is a very important subject area. In submitting your revision, make sure you adequately address all the comments by the Reviewers.

Reviewers' comments:

Reviewer's Responses to Questions

**Comments to the Author**

1. Is the manuscript technically sound, and do the data support the conclusions?

Reviewer #1: Yes

Reviewer #2: Yes

2. Has the statistical analysis been performed appropriately and rigorously? 

Reviewer #1: Yes

Reviewer #2: Yes

3. Have the authors made all data underlying the findings in their manuscript fully available?

Reviewer #1: Yes

Reviewer #2: No

4. Is the manuscript presented in an intelligible fashion and written in standard English?

Reviewer #1: No

Reviewer #2: Yes

5. Review Comments to the Author

Reviewer #1: The introduction lacks a strong motivation for the study. The paragraph placed from line 79 would have been best put as the last paragraph after a strong case has been made for the study. Lines 85 to 100 seem at best misplaced and at worst unnecessary.

The Methods section in both the abstract and the manuscript need more detail, readers need to be clear about study population, inclusion and exclusion criteria if any, study setting, data collection (clearly state that secondary data analysis was done on data collected for DHS during the period ..... through .....), variables, data analysis etc. Descriptions of how variables were named, coded and recoded that the authors put in the abstract were misplaced, these are better described under variables in the methods section.

The results section is not telling a story but rather reads like a mere description of tables. The study makes very interesting findings that are masked by the fixation on describing table, the tables must support the story being told to make for an interesting read.

Reviewer #2: Adolescent pregnancy is a very important aspect of reproductive health and this manuscript is a bold attempt to answer some pertinent questions in a country like Cameroon where studies are lacking. The effort to distinguish premarital from marital pregnancy provides more insight and data that can fine tune policies on adolescent pregnancy. There manuscript is well written but there are some areas that will need revision.

The authors carried out secondary data analysis of DHS survey data over a period of time but they did not discuss how the survey was carried out and provided very little information that can enable the assessment of the quality of the data. More details on these will be needed. Furthermore, even if detailed ethical approval was not required for the study, the authors need to discuss the ethical considerations around using the secondary data.

Lines 121-122, 234-235, 257 and 275 need to be revised.

In the results section, results that relate to the tables should be presented before moving to the figures. The authors tend to mix both, making the presentation of results incoherent in some instances.

What do they author mean by the speed of entry into marriage after pre-marital pregnancy? I understand this was derived from the time difference but care should be taken when using such descriptions. It is better to use hazard of entry consistently.

Authors should avoid using prevalence. They should use proportions consistently. The relationship between premarital pregnancy and education was not clear. In one section the author mention that the prevalence of premarital pregnancy increases with education level and in another section they state that: “intermediate levels of schooling (primary or lower secondary) recorded consistently higher probabilities of having a premarital pregnancy than those with no schooling or with the highest level of schooling”. This should be clearly articulated in the text. In the discussion section, enough explanation was not given for this. In general, for the discussion section, the authors have not used enough contemporary literature that will be relevant to the topic under consideration.

6. PLOS authors have the option to publish the peer review history of their article (what does this mean?). If published, this will include your full peer review and any attached files.

Reviewer #1: **Yes: **Brian Kumbirai Moyo

Reviewer #2: No

---

## [Author Response · Author response to Decision Letter 0]

19 Sep 2022

B. Comments by Reviewer #1

1. The introduction lacks a strong motivation for the study. The paragraph placed from line 79 would have been best put as the last paragraph after a strong case has been made for the study. Lines 85 to 100 seem at best misplaced and at worst unnecessary.

Response

The first paragraph of the introduction (Lines 49-54) was designed to provide a strong motivation for the study. We have expanded on the consequences of teenage pregnancy and motherhood. 

We have moved up the text in Lines 85-100, making the paragraph from line 79, the last paragraph of the introduction.

2. The Methods section in both the abstract and the manuscript need more detail, readers need to be clear about study population, inclusion and exclusion criteria if any, study setting, data collection (clearly state that secondary data analysis was done on data collected for DHS during the period ..... through .....), variables, data analysis etc. Descriptions of how variables were named, coded and recoded that the authors put in the abstract were misplaced, these are better described under variables in the methods section.

Response

In the first paragraph we’ve added details on how we obtained the working dataset (from the women and wealth files), and specified that we are indeed using secondary data.

We have added details on the calculation of the outcome variables and the covariates.

3. The results section is not telling a story but rather reads like a mere description of tables. The study makes very interesting findings that are masked by the fixation on describing table, the tables must support the story being told to make for an interesting read.

Response: We think this comment raises the distinction between Results and Discussion. We prefer to retain our distinction wherein the Results section is indeed a description of tables whereas, in the Discussion, we try to tell an interesting story. We hope this is acceptable to the Reviewer and the Editor.

C. Comments by Reviewer #2

1. Adolescent pregnancy is a very important aspect of reproductive health and this manuscript is a bold attempt to answer some pertinent questions in a country like Cameroon where studies are lacking. The effort to distinguish premarital from marital pregnancy provides more insight and data that can fine tune policies on adolescent pregnancy. There manuscript is well written but there are some areas that will need revision.

Response: Thanks!

2. The authors carried out secondary data analysis of DHS survey data over a period of time but they did not discuss how the survey was carried out and provided very little information that can enable the assessment of the quality of the data. More details on these will be needed. Furthermore, even if detailed ethical approval was not required for the study, the authors need to discuss the ethical considerations around using the secondary data.

Response 

In the first paragraph of the methods section, we have added an additional sentence on the DHS program. 

The methodology of DHSs is familiar and details of the Cameroon surveys can be found in the main published reports of each of the five rounds. Though the quality of DHS data is acknowledged to be high, they are, of course, far from perfect, as has been documented in many methodological reports. Of particular concern for the purposes of this paper is possible omission of recent births. Our analysis of trends in teenage pregnancy suggests somewhat poor quality of reporting in 2018. With this exception, our results are sufficiently coherent and convincing as to allay doubts about data quality.

To our knowledge, there are no major ethical considerations around using the DHS data.

3. Lines 121-122, 234-235, 257 and 275 need to be revised.

Response 

Lines 121-122: We have rephrased the beginning of sentence.

Line 257: We have removed a hanging letter ‘’t’’.

Line 275: We mistakenly have a period, where we needed a coma. 

Lines 234-235: We did not find anything that’d need to be revised. 

4. In the results section, results that relate to the tables should be presented before moving to the figures. The authors tend to mix both, making the presentation of results incoherent in some instances.

Response: We have revised the beginning of the 2nd sentence of the results section to ensure clarity from Table 1 and Figure 1. 

5. What do they author mean by the speed of entry into marriage after pre-marital pregnancy? I understand this was derived from the time difference but care should be taken when using such descriptions. It is better to use hazard of entry consistently.

Response: We have replaced speed with hazard. 

6. Authors should avoid using prevalence. They should use proportions consistently. 

Response: We have replaced ‘’prevalence of premarital pregnancy” with “likelihood of premarital agency”. 

7. The relationship between premarital pregnancy and education was not clear. In one section the author mention that the prevalence of premarital pregnancy increases with education level and in another section they state that: “intermediate levels of schooling (primary or lower secondary) recorded consistently higher probabilities of having a premarital pregnancy than those with no schooling or with the highest level of schooling”. This should be clearly articulated in the text. In the discussion section, enough explanation was not given for this. 

Response: We have expanded the text in Line 168 to clarify further. 

8. In general, for the discussion section, the authors have not used enough contemporary literature that will be relevant to the topic under consideration.

Response: We have added two additional contemporary references (now #35 and #36).

---

## [Editor Report · Decision Letter 1]

31 Oct 2022

Teenage pregnancy and timing of first marriage in Cameroon - What has changed over the last three decades, and what are the implications?

PONE-D-22-18046R1

Dear Dr. Fotso,

We’re pleased to inform you that your manuscript has been judged scientifically suitable for publication and will be formally accepted for publication once it meets all outstanding technical requirements.

Kind regards,

Grant Murewanhema, MD

Academic Editor

PLOS ONE
---

## [Editor Report · Acceptance letter]

7 Nov 2022

PONE-D-22-18046R1 

Teenage pregnancy and timing of first marriage in Cameroon - What has changed over the last three decades, and what are the implications? 

Dear Dr. Fotso:

I'm pleased to inform you that your manuscript has been deemed suitable for publication in PLOS ONE. Congratulations! Your manuscript is now with our production department. 

Kind regards, 

on behalf of

Dr. Grant Murewanhema 

Academic Editor

PLOS ONE